# The Effect of Parenteral or Oral Iron Supplementation on Fatigue, Sleep, Quality of Life and Restless Legs Syndrome in Iron-Deficient Blood Donors: A Secondary Analysis of the IronWoMan RCT

**DOI:** 10.3390/nu12051313

**Published:** 2020-05-05

**Authors:** Susanne Macher, Cornelia Herster, Magdalena Holter, Martina Moritz, Eva Maria Matzhold, Tatjana Stojakovic, Thomas R. Pieber, Peter Schlenke, Camilla Drexler, Karin Amrein

**Affiliations:** 1Department of Blood Group Serology and Transfusion Medicine, Medical University of Graz, 8036 Graz, Austria; susanne.macher@medunigraz.at (S.M.); cornelia.herster@gmx.at (C.H.); martina.moritz@klinikum-graz.at (M.M.); eva.matzhold@medunigraz.at (E.M.M.); peter.schlenke@medunigraz.at (P.S.); camilla.drexler@klinikum-graz.at (C.D.); 2Laboratory of the ÖGK, Friedrichgasse 18, 8010 Graz, Austria; 3Institute for Medical Informatics, Statistics and Documentation, Medical University of Graz, 8036 Graz, Austria; magdalena.holter@medunigraz.at; 4 Clinical Institute of Medical and Chemical Laboratory Diagnostics, University Hospital Graz, 8036 Graz, Austria; stojakovic@gmx.at; 5Division of Endocrinology and Diabetology, Medical University of Graz, 8036 Graz, Austria; thomas.pieber@medunigraz.at

**Keywords:** iron deficiency, blood donor, fatigue, RLS, iron supplementation

## Abstract

**Background**: Besides anemia, iron deficiency may cause more subtle symptoms, including the restless legs syndrome (RLS), the chronic fatigue syndrome (CFS) or sleeping disorders. **Objective**: The aim of this pre-planned secondary analysis of the IronWoMan randomized controlled trial (RCT) was to compare the frequency and severity of symptoms associated with iron deficiency before and after (intravenous or oral) iron supplementation in iron deficient blood donors. Methods/Design: Prospective, randomized, controlled, single-centre trial. (ClinicalTrials.gov: NCT01787526). Setting: Tertiary care center in Graz, Austria. Participants: 176 (138 female and 38 male) whole-blood and platelet apheresis donors aged ≥ 18 and ≤ 65 years with iron deficiency (ferritin ≤ 30ng/mL at the time of blood donation). Interventions: Intravenous iron (1 g ferric carboxymaltose, n = 86) or oral iron supplementation (10 g iron fumarate, 100 capsules, n = 90). Measurements: Clinical symptoms were evaluated by a survey before iron therapy (visit 0, V0) and after 8–12 weeks (visit 1, V1), including questions about symptoms of restless legs syndrome (RLS), chronic fatigue syndrome (CFS), sleeping disorders, quality of life and symptoms like headaches, dyspnoea, dizziness, palpitations, pica and trophic changes in fingernails or hair. Results: We found a significant improvement in the severity of symptoms for RLS, fatigue and sleep quality (*p* < 0.001). Furthermore, a significant decrease in headaches, dyspnoea, dizziness and palpitations was reported (*p* < 0.05). There was no difference between the type of iron supplementation (intravenous versus oral) and clinical outcome data. Conclusion: Iron supplementation in iron-deficient blood donors may be an effective strategy to improve symptoms related to iron deficiency and the wellbeing of blood donors.

## 1. Introduction

An imbalance between iron requirements and iron supply results in iron deficiency (ID), the most prevalent nutritive disorder worldwide [1]. It is estimated that two billion people are affected [2], and ID is the main cause for anemia besides other nutritional causes, including vitamin B12 and folate deficiency [3]. Besides anemia, ID may have other, more subtle adverse effects on an individual such as a restless legs syndrome (RLS) [4,5,6,7], chronic fatigue syndrome (CFS) [8,9,10,11] or decreased physical and cognitive performance, especially in children and premenopausal women [12,13,14]. Furthermore, ID is associated with sleeping disorders [6], trophic changes in fingernails [15] and hair [16], symptoms of pica ^(17, 18)^ and obstetric complications including low birth weight, preterm delivery and stillbirth [17].

As the largest part of total body iron is bound to hemoglobin in red cells [3], each whole blood donation (about 500 mL) results in an iron loss of 200–300 mg [18] and each apheresis donation (plasma, platelets) in a loss of 20 to 25 mg [19]. Given the maximum number of donations per year (e.g., in Austria 4–6 whole blood donations, and up to 26 platelet or 45 plasma donations), blood donations may substantially compromise the iron store of individual donors who are often iron-deficient [20,21,22,23,24,25,26,27,28,29,30,31].

In the IronWoMan trial, we randomized 176 iron-deficient blood donors (138 women, 38 men) to a single high dose of intravenous ferric carboxymaltose (1000 mg) or to oral iron (II)fumarate (100 capsules, 100 mg each) for 10 weeks. The primary endpoint was the difference in transferrin saturation between the two groups after 8–12 weeks [32]. Both median transferrin saturation and ferritin levels were significantly higher in the intravenous group (27% vs. 21%; *p* < 0.001 and 105 vs. 25 ng/mL; *p* < 0.001, respectively) while median hemoglobin levels were comparable (IV, 13.6g/dL vs. oral, 13.6 g/dL). The frequency of adverse effects was also comparable (38% in both groups) and no serious adverse events occurred.

Here, we present the results of the pre-planned secondary analysis covering the prevalence and severity of different subjective clinical symptoms related to ID before and after iron therapy (high-dose intravenous or standard oral iron).

## 2. Materials and Methods

The study was carried out according to GCP guidelines and conformed to the Declaration of Helsinki. The protocol was approved by the Austrian Agency for Health and Food Safety (AGES) and by the Ethical Committee of the Medical University of Graz. It was registered at the European Clinical Trials Database (EudraCT no. 2013-000327-14) and ClinicalTrials.gov (NCT01787526).

### 2.1. Donor Characteristics

The study population consisted of healthy male and female whole blood and platelet apheresis donors aged ≥ 18 and ≤ 65 years, fulfilling the Austrian criteria for blood donation, with ID (ferritin level ≤ 30 ng/mL) [33] but without anemia at the time of blood donation. Exclusion criteria were hemochromatosis, acute infection, pregnancy or lactation, a history of anaphylaxis to intravenous iron or other substances, and signs or symptoms suggestive of acute or chronic gastrointestinal or excessive gynecological bleeding. The study participants were randomized in a 1:1 ratio to receive either a single dose of IV ferric carboxymaltose (1 g in a short infusion, Ferinject^®^, manufacturer Vifor Pharma, Austria) or 100 capsules of oral ferrous fumarate in combination with ascorbic acid given over 8–12 weeks (each capsule containing 100 mg of bivalent iron and 20 mg ascorbic acid, Ferretab^®^ capsules, manufacturer G.L. Pharma GmbH, Austria; total dose 10 g, assuming a 10% resorption rate). Ferrous fumarate is one of the commonly used oral iron formulations in Austria.

There was no control group because we did not want to give no treatment to otherwise healthy participants with proven iatrogenic iron deficiency; most individuals were even anemic after the blood donation.

For this secondary analysis of subjective clinical endpoints, we performed a before (V0) and after (V1, 8–12 weeks after inclusion) analysis of the entire study population.

### 2.2. Study Design

The clinical trial was a prospective, randomized, controlled single-center study. Recruitment started in June 2014 and was completed in June 2016. The detailed study protocol was previously published [34]; an overview of the study procedures is given in Figure 1. In brief, ferritin screening was done from remaining serum for routine infectious disease testing in donors who had given general informed consent to participate in clinical studies. Donors with a serum ferritin ≤ 30 ng/mL were invited by telephone to participate in the study in the apheresis unit of the Department of Blood Group Serology and Transfusion Medicine. Before the first visit, a participant code was generated and a randomization for either intravenous (IV) or oral iron was performed.

During the first visit (visit 0, V0) four to five weeks after the blood donation, written informed consent was obtained, inclusion/exclusion criteria were checked and a pregnancy test in premenopausal women was performed.

A case report form was filled out by the study participants including questions about:-Symptoms of restless legs syndrome (RLS);-Symptoms of the chronic fatigue syndrome (CFS);-Sleeping behaviour;-Subjective physical, psychological health and quality of life (QOL);-Trophic changes in fingernails or hair;-Symptoms of pica.

The questions were developed based on the published criteria for RLS [35], fatigue [36], and sleeping behavior [37,38] and also included questions on symptoms known to be associated with ID, such as pica.

At visit 1 (V1), after 8 to 12 weeks, the participants completed a second questionnaire containing additional questions about adverse events and compliance with study medication. After V1, all biochemical results were summarized in a written report for each study participant and were sent by mail (including recommendations in case of specific findings).

A schematic study flowchart is given in Figure 1.


**Restless legs syndrome:**


RLS was assumed when the three specific RLS questions of the questionnaire were answered with “Yes” by the study participant. Adapted from the essential diagnostic criteria by Allen et al. [35], we asked (1) for uncomfortable sensations in the legs, accompanied by an urge to move the legs; (2) if the urge to move or unpleasant sensations begin or worsen during periods of rest and relieve by movement; and (3) if the urge to move or the unpleasant sensations is worse in the evening or night or during the day. Additionally, we asked how often the symptoms occurred.


**Fatigue:**


Based on the Multidimensional Fatigue Symptom Inventory (MFSI) [36], five different scales (general, physical, emotional, mental, vigor) were evaluated in the questionnaire.


**Quality of sleep:**


To evaluate the quality of sleep, we used the Jenkins Sleep Evaluation Questionnaire and classified the answers with the Jenkins Scale [38]. In the questionnaire, we asked how often during the previous four weeks the following symptoms occurred: (1) difficulty falling asleep, (2) waking up several times per night, (3) difficulty staying asleep, and (4) waking up feeling tired and worn out after usual amount of sleep. For every question, there was a 6-point response scale.


**Different symptoms:**


Further different questions (i.e., hair loss, brittle nails, headaches) were asked with the possibility to answer with “Yes” or “No”.

The study participants subjectively indicated their quality of life with five differences on a Likert type style scale, ranging in five steps between “very poor” and “very good”.

### 2.3. Statistical Analysis

This analysis followed the intention-to-treat principle. Descriptive statistics of the data are presented as median and quartile or absolute and relative frequencies, depending on the type of data. Differences between V0 and V1 are analyzed either by the Wilcoxon signed-rank test or by the Mc Nemar’s test. The significance level was set to alpha = 0.05. SPSS statistical software (version 25.0; IBM Corp, Armonk, NY, USA) was used to perform the statistical data analysis. Since the questionnaires used in this study were translated by the study team, their structure was examined before further use. The usability of each questionnaire was evaluated by factorial structure (confirmatory factor analysis) and fit indices. Only questionnaires which met the criteria were analyzed. These analyses were performed with R software (version 3.5.0), using the package “Multidimensional Item Response Theory, mirt” [39].

## 3. Results

Between June 2014 and June 2016, a total of 467 blood donors (370 women, 97 men) with a serum ferritin ≤ 30 ng/mL were invited to participate in the study. A total of 291 declined to participate and 176 were included (138 women, 78%, 38 men, 22%). The total study duration for each participant was 8 to 12 weeks. In August 2016, the last participant completed the study. Demographic data of the study participants are presented in Table 1.

The primary hypothesis of the study was that intravenous iron is superior to oral iron in improving iron stores. The primary endpoint of the study was the difference in the transferrin saturation after iron therapy between the two treatment groups. All the biochemical data have been published previously (32). In brief, the primary endpoint, transferrin saturation, was significantly higher after IV iron (27 (23–35)%, vs. 21 (16–32)% compared to PO; *p* < 0.001), as was the ferritin level (IV 105 (75–145) vs. PO 25 (17–34) ng/mL; *p* < 0.001). Anemia was present in 75% of all donors at V0. After IV or oral iron, hemoglobin levels were similar (IV 13.6 (13.0–14.4) vs. PO 13.6 (13.0–14.2) g/dL). Four patients (all in the oral iron group) were lost to follow-up. Participants recommended both treatments (IV 91% vs. PO 79%) and adherence was excellent in both groups (IV 100%, PO 95% who took >90 capsules). The frequency of adverse effects was comparable (38% in both groups) and no serious adverse events occurred [31].

In this pre-panned secondary analysis we compared the clinical outcome of the study participants before and after iron therapy.

### 3.1. Clinical Outcomes

A summary of the reported clinical outcomes (restless legs syndrome, fatigue, sleep and a number of different symptoms) is given in Table 2.

#### 3.1.1. Restless Legs Syndrome

At baseline, 19.9 % (31/156) and after iron therapy, 17.1% (26/152) of the participants fulfilled the diagnostic criteria for having a RLS. Eighteen study participants fulfilled the diagnostic criteria at both visits, and 13 participants fulfilled the criteria only at V0 and not at V1.

Comparing the frequency of symptoms of the 18 study participants plus the 13 participants with RLS only at V0, a significant improvement in RLS symptoms between baseline and after IV or oral iron was seen (symptom score at V0: min. two, max. seven, median four versus at V1: min. 0, max. six, median two; *p* < 0.001).

Detailed results are shown in Table 2 and Figure 2.

#### 3.1.2. Fatigue

Each of the five scales (general, physical, emotional, mental, vigor) and the global evaluation for fatigue improved significantly after iron therapy (*p* < 0.001). Results are shown in Table 2 and Figure 3.

#### 3.1.3. Quality of Sleep

For all four qualities of sleep, there was a statistically significant improvement after iron therapy (*p* < 0.05). Results are shown in Table 2 and Figure 4.

#### 3.1.4. Different Symptoms

The presence of different symptoms possibly related to ID at the two visits (brittle nails, hair loss, pica, etc.) is shown in Table 2 and Figure 5.

Between V0 and V1, no difference was seen for the subjective assessment of quality of life (Figure 6).

#### 3.1.5. Further Results–Type of Iron Treatment and Ferritin Levels at Visit 1

There was no significant difference between IV and oral iron therapy for all reported outcomes (data not shown). There was also no significant difference between subgroups with lower or higher ferritin levels at visit 1 (data not shown).

## 4. Discussion

In this pre-planned secondary analysis of the IronWoMan RCT [32], we aimed to evaluate if clinical symptoms associated with ID improve after iron supplementation. Overall, we are able to give an overview on subjectively relevant clinical symptoms supporting the concept that ID is associated with a complex variety of subtle adverse effects in addition to anemia, and that these symptoms respond to iron replacement therapy [40]. This is not surprising, as iron is an essential micronutrient involved in many physiological processes such as oxygen transport and utilization, oxidative phosphorylation, mitochondrial function, DNA biosynthesis and ATP production [41,42].

Our findings are particularly relevant for blood donors who are otherwise healthy and expose themselves to a high risk of iatrogenic non-anemic ID, especially after repeated blood donations.

### 4.1. Restless Legs Syndrome (RLS)

RLS is one of the most prevalent neurologic diseases [43]. An association between RLS and ID has been proposed and iron treatment is regularly prescribed in RLS. Magnetic resonance imaging studies revealed a deficiency of iron in different areas of the brain [44] and this was confirmed by direct neuro-pathologic demonstration of low levels of iron in the substantia nigra of affected patients [45]. A relation between low serum ferritin and symptoms of RLS [5,6] and an improvement in RLS symptoms with iron treatment has also been demonstrated [4,46]. Earley et al. [4] found an RLS remission rate of 60% after intravenous iron treatment and Wang et al. (46) demonstrated a mean decrease in the International Restless Legs Scale score of 10.3 in the iron group versus 1.1 in the placebo group (*p* = 0.01).

A recently published review concluded that of all known parameters associated with RLS, including dopamine, glutamate, serotonin and ferritin levels, the most consistent finding appears to be ID [7]. Thus, iron replacement is suggested for RLS patients with a ferritin level < 75 ng/mL [47,48] and, according to recent clinical practice guidelines, intravenous iron is even effective for the treatment of moderate to severe RLS in patients with ferritin levels <300 ng/mL [49].

In our study population of relatively young and healthy but iron-deficient blood donors, the frequency of RLS at V0 was >20%, which is higher than the reported prevalence of RLS between 5% and 15% in the general population [50,51,52]. This high prevalence in blood donors is comparable to the numbers reported by others (Birgegard et al.: 18%, Ulfberg et al: 18%, Burchell et al.: 17%) [30,53,54]. We also compared the severity of self-reported RLS symptoms at baseline (V0) and after iron therapy and found a significant improvement after iron supplementation (*p* = 0.001). These results are consistent with the findings of a double-blind, placebo-controlled study by Wang et al. where a statistically significant improvement in RLS patients with low serum ferritin levels (15–75 ng/mL) after oral iron treatment was demonstrated [46].

### 4.2. Fatigue

All domains of reported fatigue symptoms in our study indicated a significant improvement after iron treatment (*p* < 0.001). In a Swiss double blind randomized controlled trial in a primary care setting, oral iron was found to reduce fatigue scores after one month of treatment in women with a baseline ferritin level of < 50 ng/mL [9].

In contrast, two other Swiss randomized controlled trials in blood donors did not show a clinical benefit by either oral or intravenous iron supplementation. However, Waldvogel et al. supplemented 80 mg oral iron per day for only 4 weeks and Fontana et al. gave 800 mg intravenous iron once, but no detailed results have yet been published [55,56]. In line with our results, a recent review including 18 trials enrolling 1170 non–anemic iron deficient adults found that iron supplementation led to a reduction in fatigue [10]. Two other meta-analyses also reported that iron appears to be effective in reducing fatigue in iron-deficient patients without anemia [40,57].

An interesting recent study in heart failure patients demonstrated more severe myopathy in ID and greater acidosis with exercise, giving a plausible pathophysiological explanation for muscle fatigue, which is likely also relevant for healthier individuals [58].

### 4.3. Quality of Sleep

Sleep disorders often arise as a consequence of, or in combination with, RLS [5,6,43,53,59], but the available data do not appear to be as convincing as they are for RLS. While Kryger et al. [5] found an improvement in sleep quality in iron-deficient patients after iron therapy, in the study by Cho et al. [59] the sleep quality after iron therapy compared to a placebo group did not significantly improve (sleep quality index of 11.3 in the iron group versus 10.9 in the placebo group). In our study, we evaluated the quality of sleep with the Jenkins Sleep Evaluation Questionnaire and classified the answers with the Jenkins Scale [38]. All four items—difficulty falling asleep, waking up several times per night, difficulty staying asleep, and waking up feeling tired and worn out after usual amount of sleep—showed a clear and significant improvement after iron therapy.

### 4.4. Other Symptoms

Unspecific symptoms including pica, brittle nails, hair loss and a plain tongue were reported more often before than after iron therapy (statistically nonsignificant). In the literature, the association of ID with these symptoms also remains controversial [15,16,60,61].

Pica is not specific to ID [62,63]. However, pica for ice responds rapidly to treatment with iron (63). Bryant et al. found a significantly higher percentage of pica among blood donors with ID (11%) compared to iron-sufficient donors (4%). Furthermore, pica symptoms resolved completely with oral iron supplements [21]. In our analysis, we found a nonsignificant improvement after iron supplementation, but only 6% of participants reported pica before and 3% after iron therapy.

Headache, dyspnea and dizziness improved significantly in our study. However, this may also be explained by the high percentage of anemic blood donors after the last blood donation and before iron therapy (73% of women and 82% of men) with a mean Hb of 11.7 g/dl and with the higher mean Hb of 13.6 g/dl after iron supplementation [32]. Interestingly, there was no difference between IV and oral iron therapy and the attained ferritin concentration at visit 1, although our study was certainly underpowered for such analyses.

### 4.5. Strengths and Limitations

The strengths of our study are the selective inclusion of iron-deficient blood donors, the excellent compliance and the low number of participants lost to follow up.

However, our study has some limitations: first, it was performed in a single European academic institution. Second, the follow up was limited to 3 months. Third, questionnaires are subjective and fourth, and most importantly, we did not include a placebo group. Therefore, we were unable to assess the natural course of current clinical practice (not using any iron supplements) and the influence of the hemoglobin increase per se. However, the very high incidence of iatrogenic anemia at baseline (75%) and a recent RCT [64] suggest that the recovery to pre-donation hemoglobin concentrations and iron stores typically takes substantially longer than the whole blood inter-donation interval.

## 5. Conclusions

Our findings support the hypothesis that side effects of ID are independent of anemia and can cause clinically relevant subjective symptoms in otherwise healthy, iron-deficient blood donors. These subtle symptoms—that usually go unnoticed if not actively asked for—are clearly improved by iron treatment.

Our study was relatively small for the evaluation of clinical endpoints. The fact that we were able to see a significant improvement in several of the analyzed clinical symptoms suggests that the benefits outweigh potential risks of iron treatment and are of clinical relevance. Interestingly, the approach of iron substitution (oral or IV) was not crucial for the improvement in symptoms. Therefore, we suggest that blood donation services should identify donors at risk for iatrogenic ID and offer iron supplementation accompanied by the extension of the donation interval to maintain the health of each blood donor, especially in premenopausal women, the highest risk group.

## Figures and Tables

**Figure 1 nutrients-12-01313-f001:**
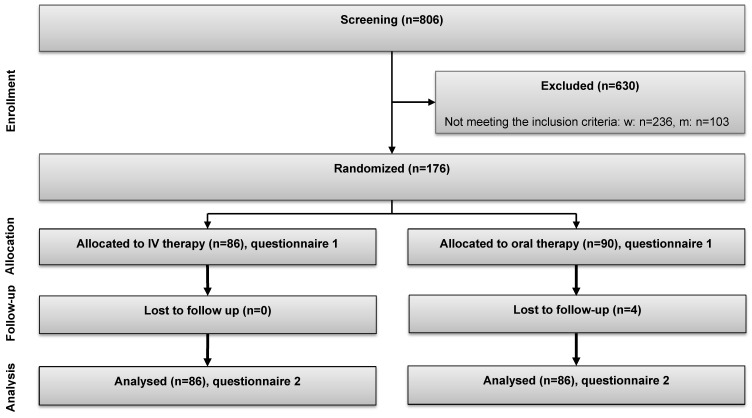
Study flowchart.

**Figure 2 nutrients-12-01313-f002:**
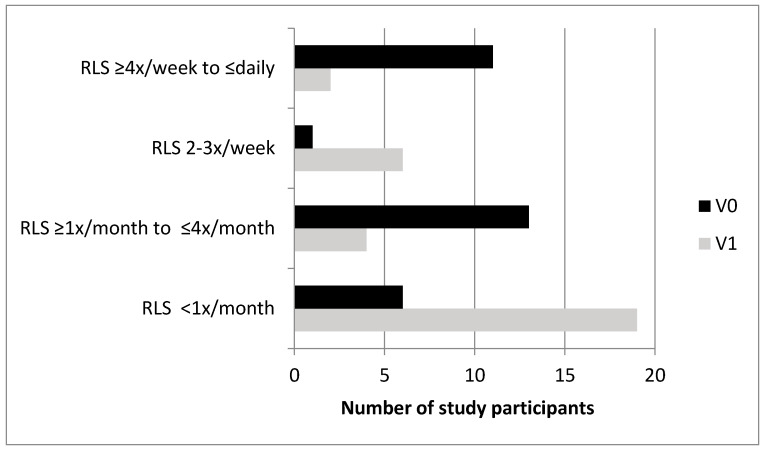
Bar diagram showing the number of study participants (x-axis) who indicate the frequency of their restless legs syndrome (RLS) at the two time points before (V0, dark bars) and after IV or oral iron therapy (V1, bright bars) on the y-axis. The decrease in the RLS occurrence between V0 and V1 was statistically significant (*p* < 0.001, n = 31).

**Figure 3 nutrients-12-01313-f003:**
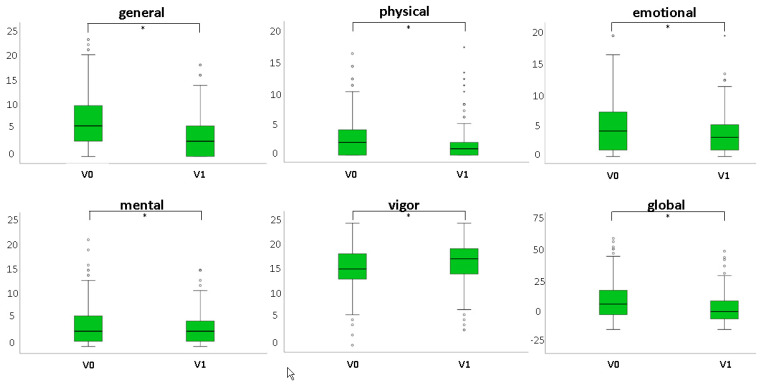
Comparison of median and 1^st^/3^rd^ quartiles for the fatigue scales before (V0) and after (V1) IV or oral iron therapy. For the 4 scales general, physical, emotional and mental a lower level indicates an improvement, for the scale vigor a higher level indicates an improvement (* *p* < 0.001 for all, n = 160).

**Figure 4 nutrients-12-01313-f004:**
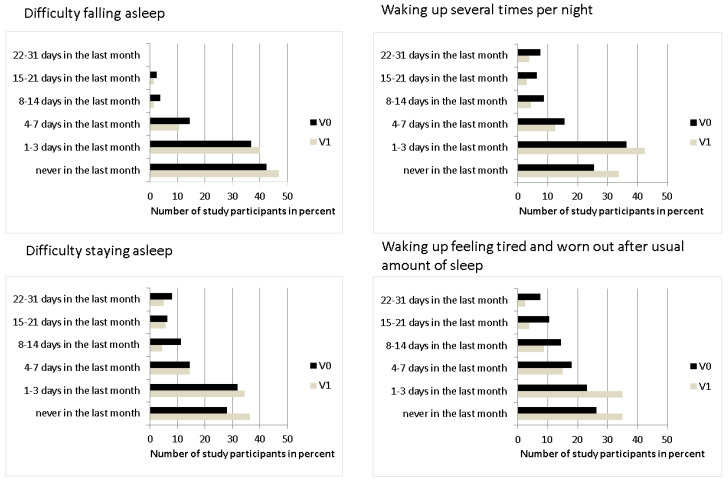
Bar diagrams indicating the percentage of study participants on the x-axis with the different response alternatives according to the Jenkins scale for sleeping behavior on the y-axis before (V0, dark bars) and after iron therapy (V1, bright bars). For all 4 quality of sleep items, there was a statistically significant improvement between V0 and V1 (*p* < 0.05 for all, n = 160).

**Figure 5 nutrients-12-01313-f005:**
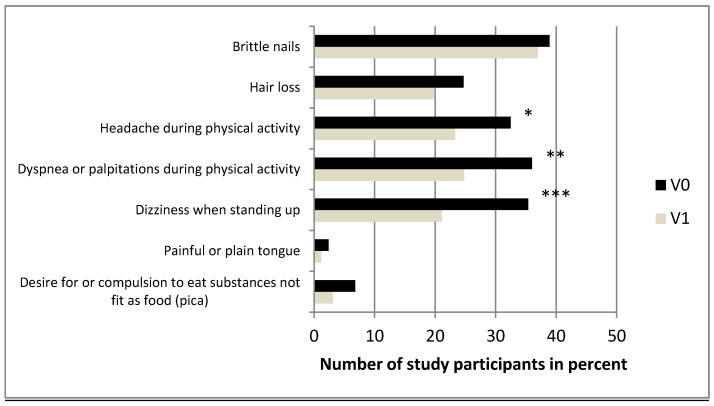
Bar diagram indicating the percentage of study participants on the x-axis with the appropriate symptoms indicated on the y-axis before (V0, dark bars) and after iron therapy (V1, bright bars). (* *p* = 0.02, ** *p* = 0.005, *** *p* < 0.001, n = 161–165).

**Figure 6 nutrients-12-01313-f006:**
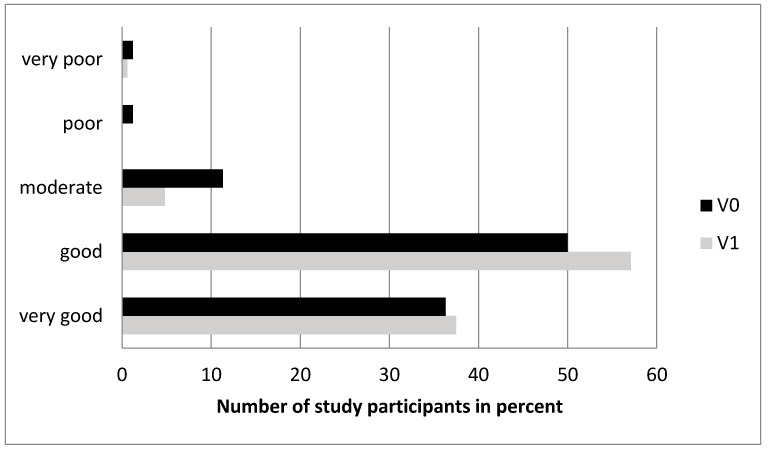
Bar diagram indicating the percentage of study participants (x-axis) rating the quality of their life between very poor and very good (y-axis) before (V0, dark bars) and after iron therapy (V1, bright bars). (no statstically significant differences, n = 168).

**Table 1 nutrients-12-01313-t001:** Basic characteristics of the study participants (IV, study group with intravenous iron; PO, study group with oral iron; SD standard deviation).

Type of Donor	Women	Men	Total
Whole blood	126	31	157
Platelet (apheresis)	12	7	19
**Study Group**			
IV	69	17	86
PO	69	21	90
Sum	138	38	176
**Donor Characteristics**			
Age, mean (SD), years	36 (13)	46 (13)	
Weight, mean (SD), kg	69 (11)	82 (16)	
Body mass index, mean (SD)	24 (4)	26 (4)	
Total calculated blood volume, mean (SD), liters	4.0 (0.3)	5.3 (0.6)	

**Table 2 nutrients-12-01313-t002:** Clinical outcome data at V0 and V1. Nominal data are presented as N (%), ordinal data as Median (IQR).

		N	V0	V1	P
**Restless Legs Syndrome**	Diagnosis	134	31 (23.1%)	23 (17.2%)	-
	Severity		4 (3–6)	2 (1–5)	<0.001
**Fatigue**	General	169	6 (3–10)	3 (0–6)	<0.001
	Physical		2 (0–4)	1 (0–2)	<0.001
	Emotional		4 (1–7)	3 (1–5)	<0.001
	Mental		3 (1–6)	3 (1–5)	<0.001
	Vigor		15 (13–18)	17 (14–19)	<0.001
	Overall		0 (−10–13)	−7 (−14–3)	<0.001
**Quality of Sleep**	Difficulty falling asleep	160	5 (5–6)	5 (5–6)	0.004
	Waking up several times		5 (4–5)	5 (5–6)	<0.001
	Difficulty staying asleep		5 (3–6)	5 (4–6)	0.002
	Waking up tired		4 (3–6)	5 (4–6)	<0.001
**Different Symptoms**	Brittle Nails	162	63 (38.9%)	60 (37%)	0.678
	Hair loss		40 (24.7 %)	32 (19.8%)	0.096
	Headache		53 (32.5%)	38 (23.3%)	0.02
	Dyspnea		58 (36.0%)	40 (24.8%)	0.005
	Dizziness		57 (35.4%)	34 (21.1%)	<0.001
	Painful or plain tongue		4 (2.4%)	2 (1.2%)	0.687
	Pica		11 (6.8%)	5 (3.1%)	0.109
**Quality of Life**		171	4 (4–5)	4 (4–5)	-

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
