# Peer review of "The Effect of Parenteral or Oral Iron Supplementation on Fatigue, Sleep, Quality of Life and Restless Legs Syndrome in Iron-Deficient Blood Donors: A Secondary Analysis of the IronWoMan RCT"

_nutrients, 2020, doi:10.3390/nu12051313_

Round 1

Reviewer 1 Report

Please move the analysis of the groups in terms od demographical features to materials and methods section. Did they sign informed consent ?

Are there any difference determined by sex criteria ? 

Were there any differences in efficacy between the oral and IV treatment in those conditions ?

The study would benefit highly if the control group ws included in the analysis.

Author Response

Dear editor,

Dear reviewers,

We would like to thank the reviewers and editor for the rapid and constructive review of our manuscript. We aimed to clarify all raised topics in a point-by-point manner as outlined below.

Comments point by point

Please move the analysis of the groups in terms od demographical features to materials and methods section.

Response (R): Thank you very much for your appreciation of our work and for your valuable input. We changed the Material and Methods Section to make the study design clearer.

Did they sign informed consent ?

(R): Yes, certainly all study participants signed written informed consent. This is described in 2.2. Study design in the Materials and Methods section.

Are there any difference determined by sex criteria ? 

(R): No, there were no relevant or significant differences regarding to sex. However, we are afraid our sample size was too small to exclude a small effect.

Were there any differences in efficacy between the oral and IV treatment in those conditions ?

(R): This is a very relevant question, but again - no, there were no significant differences regarding to the type of treatment, this was clarified in the Methods section and the end of the Introduction section. However, we are afraid our sample size is too small to exclude a relevant effect in clinical routine.

The study would benefit highly if the control group ws included in the analysis.

(R): Yes we absolutely agree. The addition of a control group would have been helpful. Unfortunately, due to restricted resources we were unable to include a control group. Also, we found it ethically challenging to include a group with verified iatrogenic iron defiiciency and not treat it so decided against it after extensive discussions. We also added this to the Methods section.

Reviewer 2 Report

Comments point by point

Introduction

The authors give a definition of RLS, it is not necessary (people who are reading the paper are supposed to know RLS) and they do not give a definition of CFS.

The introduction is clear and concise

Material and methods

iron deficiency (ferritin level ≤30 ng/ml) : give a reference of this definition of iron deficiency, that is the focus of the study

without anemia : what cut off + 1 reference

Except for RLS symptom evaluation is based on a validated scale (sleep, CFS), for other symptoms the possibility to answer with “Yes” or “No” sounds pretty pragmatic. There are many quality of life scales, why did the authors used a Lickert type scale?

It is a randomized study: the working hypothesis is that intravenous iron is superior to oral iron therefore what is the number of patients to be included?

Results

First remark: the working hypothesis is that intravenous iron is superior to oral iron but there is no separate data for each group, results are presented as if the purpose of the study was to compare pre and post iron (IV and oral)

Second remark: why didn’t the authors perform the analysis for the RLS endpoint with the whole population? When reading the results the results between inclusion 19.9 % (31/156) and after iron therapy, 17.1% (26/152), there doesn't seem to be any difference.

A significant improvement of RLS symptoms between baseline and after IV or oral iron was determined (p<0.001): yes but for which end point. RLS ≥4x/week to ≤daily: yes? RLS 2-3x/week: no? RLS ≥1x/month to ≤4x/month: yes? RLS <1x/month: no? The results are not clear from Figure 2.

Bradly speaking, the results would be more legible in the form of two tables: IV vs oral and pre and post iron with numerical data and statistical analysis for each parameter.

Discussion

Besides anemia, iron deficiency may have more subtle adverse effects. Yes but give some examples: heart failure, stroke…

RLS: there is no result of previous studies about iron in RLS treatment

However, Waldvogel et al. supplemented 80 mg oral iron per day for only 4 weeks and Fontana et al. gave 800 mg intravenous iron once but no detailed results have yet been published. Is it true? The results of Waldvogel is: No significant clinical effect was observed for fatigue (-0.15 points, 95% confidence interval -0.9 points to 0.6 points, P = 0.697) or for other outcomes. I did not find the reference for Fontana. At the opposite one important reference is lacking for fatigue:

Verdon F, Burnand B, Stubi CL, Bonard C, Graff M, Michaud A, Bischoff T, de Vevey M, Studer JP, Herzig L, Chapuis C, Tissot J, Pécoud A, Favrat B: Iron supplementation for unexplained fatigue in non-anaemic women: double blind randomised placebo controlled trial. BMJ. 2003, 326 (7399): 1124-10.1136/bmj.326.7399.1124.

Quality of sleep. Our results support previous findings that in iron deficient individuals with sleep disorders, iron therapy improves symptoms. Yes but give some data.

General comments

There are too many methodological errors: the working hypothesis is that intravenous iron is superior to oral iron but no calculation of the number of patients to be included and results reporting as if the purpose of the study was to compare pre and post iron (IV and oral).

The data is too difficult to read and to understand. It would be easier to have data for RLS, fatigue and sleep since other symptoms are not assessed with validated scales.

In the discussion an essential point is not developed in the discussion: side effects of iron deficiency are independent of anemia. Furthermore, the discussion could have focused on  RLS, sleep and fatigue with more references about previous works.

Author Response

We would like to thank the editor and the three reviewers for the rapid and constructive review that helped us to improve our manuscript. We aimed to clarify all raised topics in a point-by-point manner as outlined below. In the revised manuscript, these changes are highlighted in yellow.

Comments point by point

Introduction

The authors give a definition of RLS, it is not necessary (people who are reading the paper are supposed to know RLS) and they do not give a definition of CFS.

(R): Thank you for pointing this out. We have deleted the definition of RLS.

The introduction is clear and concise

Material and methods

iron deficiency (ferritin level ≤30 ng/ml) : give a reference of this definition of iron deficiency, that is the focus of the study

( R ) Thank you for your comment: We have included a recent reference by Schrage et al, where the ferritin level of ≤30 ng/ml is defined as absolute iron deficiency, common in internal medicine, and associated with adverse outcomes:

Schrage, B., Rübsamen, N., Schulz, A. et al. Iron deficiency is a common disorder in general population and independently predicts all-cause mortality: results from the Gutenberg Health Study. Clin Res Cardiol (2020). https://doi.org/10.1007/s00392-020-01631-

Except for RLS symptom evaluation is based on a validated scale (sleep, CFS), for other symptoms the possibility to answer with “Yes” or “No” sounds pretty pragmatic. There are many quality of life scales, why did the authors used a Lickert type scale?

(R): Thank you for your comment. We used this scale only for pragmatic reasons – the questionnaires the participants were asked to complete were already 7 pages long, and typical QoL scores such as the SF-36 are also several pages long so we decided to try to collect some subjective data while still keeping the time needed for the participants acceptable.

It is a randomized study: the working hypothesis is that intravenous iron is superior to oral iron therefore what is the number of patients to be included?

(R): We apologize if it was not entirely clear that this was a RCT with primary endpoint transferrrin saturation. The sample size calculation was done for the primary endpoint to detect a difference in mean of 8% in transferrin saturation between the high-dose intravenous iron group and the oral iron group. The current manuscript evaluates the secondary subjective outcomes where no formal sample size analysis was performed.

Results

First remark: the working hypothesis is that intravenous iron is superior to oral iron but there is no separate data for each group, results are presented as if the purpose of the study was to compare pre and post iron (IV and oral)

(R): This is a very relevant question and valuable thought. The working hypothesis is correct for the primary analysis (biochemical response, published in Clinical Nutrition 2020). In this exploratory analysis of the subjective clinical data, we focused on the comparison pre and post iron therapy, as you pointed out. We tried to clarify this better at the end of the Introduction section. See also response to R1.

Second remark: why didn’t the authors perform the analysis for the RLS endpoint with the whole population? When reading the results the results between inclusion 19.9 % (31/156) and after iron therapy, 17.1% (26/152), there doesn't seem to be any difference.

(R) The analysis was only meaningful for the participants who fulfilled the definition for RLS at V0. Not the amount of participants with RLS at V0 vs. V1 is compared, but the severity of symptoms. Unfortunately only 156/176 study participants completed the RLS questions at V0 and 152/176 at V1. So we evaluated only the CRFs we had, and indeed there was no difference between the number of participants with RLS symptoms before and after iron therapy. Additionally the participants indicated the frequency of their RLS symptoms, and comparing these an improvement could be seen.

A significant improvement of RLS symptoms between baseline and after IV or oral iron was determined (p<0.001): yes but for which end point. RLS ≥4x/week to ≤daily: yes? RLS 2-3x/week: no? RLS ≥1x/month to ≤4x/month: yes? RLS <1x/month: no? The results are not clear from Figure 2.

RLS symptoms were compared with a paired Wilcoxon-Test, since the severity of symptoms is ordinal scaled. Severity of symptoms overall were compared V0 against V1 and not the individual categories against each other. This sentence with more detail was added:

a significant improvement of RLS symptoms between baseline and after IV or oral iron was seen (symptom score at V0: min. 2, max. 7, median 4 versus at V1: min. 0, max. 6, median 2; p<0.001).

Broadly speaking, the results would be more legible in the form of two tables: IV vs oral and pre and post iron with numerical data and statistical analysis for each parameter.

(R): Thank you. We have added a summary table (table 2) on the effect on the dfference between the time points V0 and V1 and agree that this adds to a clearer picture. IV vs oral iron did not produce different results.

Discussion

Besides anemia, iron deficiency may have more subtle adverse effects. Yes but give some examples: heart failure, stroke…

(R): Thank you. We have added the appropriate references.

We described the subtle adverse effects in the Introduction section. We also included appropriate references but did not want to be redundant in the Discussion section.

RLS: there is no result of previous studies about iron in RLS treatment

(R): Thank you. We have added and commented more references relevant to this topic in the discussion:

Earley et al. (4) found a RLS remission rate of 60% after intravenous iron treatment and Wang et al. (43) demonstrated a mean decrease of the International Restless Legs Scale score of 10.3 in the iron group versus 1.1 in the placebo group (p=0.01).

However, Waldvogel et al. supplemented 80 mg oral iron per day for only 4 weeks and Fontana et al. gave 800 mg intravenous iron once but no detailed results have yet been published. Is it true? The results of Waldvogel is: No significant clinical effect was observed for fatigue (-0.15 points, 95% confidence interval -0.9 points to 0.6 points, P = 0.697) or for other outcomes. I did not find the reference for Fontana. At the opposite one important reference is lacking for fatigue:

Verdon F, Burnand B, Stubi CL, Bonard C, Graff M, Michaud A, Bischoff T, de Vevey M, Studer JP, Herzig L, Chapuis C, Tissot J, Pécoud A, Favrat B: Iron supplementation for unexplained fatigue in non-anaemic women: double blind randomised placebo controlled trial. BMJ. 2003, 326 (7399): 1124-10.1136/bmj.326.7399.1124.

(R): Thank you. Up to April 20, indeed, the Fontana study had only been published as an abstract. The Verdon reference (reference 9) is now also used in the Discussion:

In a Swiss double blind randomized controlled trial in a primary care setting, oral iron was found to reduce fatigue scores after one month of treatment in women with a baseline ferritin level of < 50 ng/ml (9).

Quality of sleep. Our results support previous findings that in iron deficient individuals with sleep disorders, iron therapy improves symptoms. Yes but give some data.

(R): Thank you. We have added and commented appropriate references in the discussion.

General comments

There are too many methodological errors: the working hypothesis is that intravenous iron is superior to oral iron but no calculation of the number of patients to be included and results reporting as if the purpose of the study was to compare pre and post iron (IV and oral).

(R): See also above - we apologize if it was not entirely clear that this was a RCT with a laboratory primary endpoint (transferrrin saturation) that has recently been published in Clinical Nutrition. Therefore, no power calculation for the secondary endpoints presented in this paper was performed.

The data is too difficult to read and to understand. It would be easier to have data for RLS, fatigue and sleep since other symptoms are not assessed with validated scales.

(R): Thank you for this comment. We added Table 2 and tried to make the results clearer throughout the manuscript. Although other reported symptoms were not assessed with validated questionnaires (we are not aware of any covering all these typical ID symptoms), we would still prefer to keep these informations in the manuscript because we believe they may be relevant for counseling in clinical routine

In the discussion an essential point is not developed in the discussion: side effects of iron deficiency are independent of anemia. Furthermore, the discussion could have focused on  RLS, sleep and fatigue with more references about previous works.

(R): Thank you. We have amended the discussion as suggested and included new relevant references.

Reviewer 3 Report

THE EFFECT OF PARENTERAL OR ORAL IRON SUPPLEMENTATION ON FATIGUE, SLEEP, QUALITY OF LIFE AND RESTLESS LEGS SYNDROME IN IRON DEFICIENT BLOOD DONORS: A SECONDARY ANALYSIS OF THE IRONWOMAN RCT is a well though-out study and a well written manuscript. I have a few suggestions for the authors

  1. The authors should include more details about the IronWoman trail in the introduction
  2. Authors should specify the rationale/advantages/disadvantages for the use of the Fe source used in this study (since ferrous sulfate is the common salt used in iron supplements).
  3. Please specify the rationale for the hypothesis in the manuscript.
  4. There are no error bars on most of the graphs presented in the study which brings the credibility of the significance of the data into question.
  5. WHile the authors show an improvement in several parameters upon Fe supplementation, it would greatly strengthen the manuscript if the authors could provide the understanding or the reasons (biochemical or clinical) for such improvements to the readers.

Author Response

We would like to thank the editor and the three reviewers for the rapid and constructive review of our manuscript. We aimed to clarify all raised topics in a point-by-point manner as outlined below. In the revised manuscript, these changes are highlighted in yellow.

  1. The authors should include more details about the IronWoman trail in the introduction

(R): Thank you for the thoughtful review of our mansucript. We agree – we have added more details on the IronWoMan study in the introduction.

  1. Authors should specify the rationale/advantages/disadvantages for the use of the Fe source used in this study (since ferrous sulfate is the common salt used in iron supplements).

(R): Thank you. Ferrous fumarate is very common in Austria besides ferrous sulfate. We have added this information in the Methods section.

  1. Please specify the rationale for the hypothesis in the manuscript.

(R): Thank you. We have added more reasoning on the hypothesis including some relevant references.

  1. There are no error bars on most of the graphs presented in the study which brings the credibility of the significance of the data into question.

(R): Thank you for this comment. We did not add error bars to the boxplots since the variation in the data is seen in the whiskers of the boxplot figures. Adding an error bar which is based on the mean and standard deviation would not provide additional meaningful information but make the figures busier.

  1. WHile the authors show an improvement in several parameters upon Fe supplementation, it would greatly strengthen the manuscript if the authors could provide the understanding or the reasons (biochemical or clinical) for such improvements to the readers.

(R): We agree. This coincides with Reviewer 2’s comment - we have amended the discussion as suggested and included relevant references.

Round 2

Reviewer 1 Report

The manuscript is improved and is ready for publication.

Reviewer 3 Report

The modified manuscript looks good.